# Gram-scale selective telomerization of isoprene and $CO_2$ toward 100% renewable materials

Marius D. R. Lutz ⓘ, Felix Kracht, Kota Marumoto & Kyoko Nozaki ⓘ ✉

Carbon dioxide ($CO_2$) is an ideal chemical feedstock due to its abundance, low cost, low toxicity and its role as a greenhouse gas. Telomerization with butadiene give rise to functional small molecules and polymers with significant $CO_2$ content, but the fossil origin of the olefin offsets sustainability benefits. Here, we present a palladium-catalyzed telomerization of $CO_2$ with isoprene, two of the most prevalent organic compounds in the atmosphere, yielding "COOIL", an ideally 100% renewable δ-lactone containing 24 wt% $CO_2$, with high selectivity and turnover numbers above 100. A combination of a Pd catalyst, acetate, and controlled water promoted selectivity and conversion. Density functional theory calculations reveal reductive elimination as the rate-limiting and selectivity-determining step, preceded by isoprene dimerization. The head-tail pathway is the kinetic pathway while the tail-tail product is the thermodynamic product. This functionalized lactone also shows promise for polymerization under Lewis acid-promoted conditions, opening avenues for sustainable polymers from $CO_2$ and bio-derived feedstocks.

Developing methods for the chemical fixation of carbon dioxide ($CO_2$) offers significant potential to reduce its atmospheric levels, addressing a key contributor to climate change. This approach not only utilizes an abundant and inexpensive feedstock but also creates a $CO_2$-negative process[1]. Despite $CO_2$ being an ideal C1 feedstock due to its global availability, low cost, and low toxicity, its use in organic synthesis is challenged by its low reactivity and high oxidation state[2].

3-Ethylidene-6-vinyltetrahydro-2H-pyran-2-one (EVP), a six-membered lactone synthesized from two butadiene units and $CO_2$, has gained attention as a versatile monomer for polymers with high $CO_2$ content (Fig. 1a)[3,4]. EVP is accessible through an atom-economical palladium-catalyzed telomerization of butadiene under a $CO_2$ atmosphere[5,6]. In the past decade, EVP's potential as a monomer for sustainable $CO_2$-based polymers has been extensively explored[7]. Various polymerization procedures have been investigated, including free radical polymerization, controlled radical polymerization[8], coordination-insertion polymerization[9], ring-opening polymerization[10-14], amongst others[2-4]. In parallel, improved homogeneous and heterogeneous protocols have been developed to access EVP in high yields (Fig. 1b)[7,15-19].

Isoprene, a C5 hydrocarbon, is the most abundantly emitted hydrocarbon in the atmosphere after methane[20,21]. It is estimated that emissions release 400–500 teragrams of carbon (Tg C) from plants per year[22,23], approximately matching the amount of biogenic methane emissions[22,24]. In contrast to butadiene, which is derived from fossil resources, isoprene is a feedstock with the inherent potential for renewable sourcing independent of petrochemical processes, albeit current production routes from biomass are not competitive in terms of energy utilization[25]. Despite its abundance and the versatility of its 1,3-diene motive, isoprene's use as a renewable carbon source is largely focused on natural rubber and adhesive production[26-28]. This under-utilization of such an abundant resource underscores the need to explore its potential for sustainable polymer production. Sequestering the two ubiquitous climate gases, $CO_2$ and isoprene, into an EVP analog and its derived polymers could be a transformative step towards a more sustainable polymer economy.

Department of Chemistry and Biotechnology, Graduate School of Engineering, The University of Tokyo, Bunkyo-ku, Tokyo 113-8656, Japan.
✉e-mail: nozaki@chembio.t.u-tokyo.ac.jp

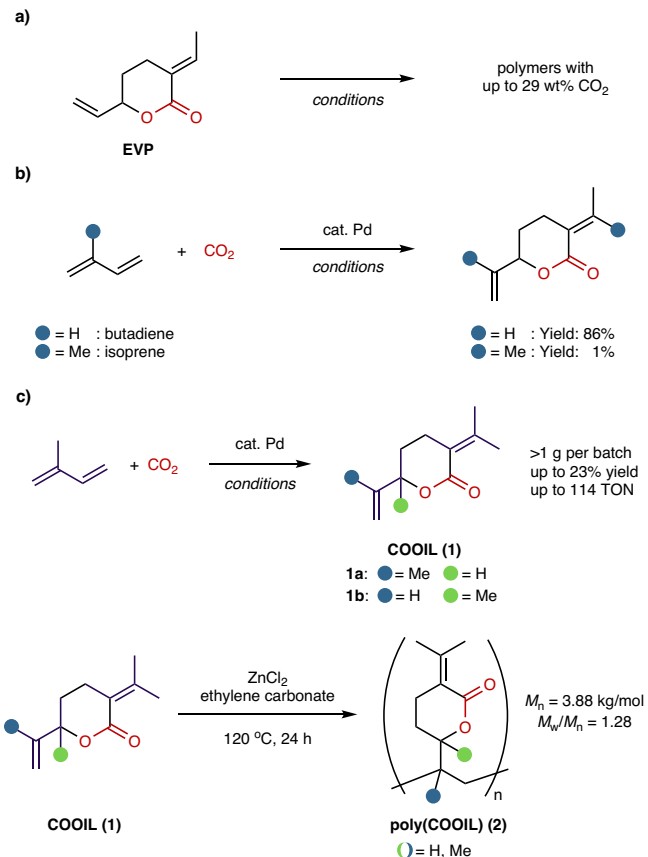

**Fig. 1 | Telomerization of 1,3-dienes and CO₂ towards renewable lactone building blocks. a** Polymerization of EVP derived from butadiene and CO₂.
**b** Comparison of butadiene and isoprene in the telomerization with CO₂.
**c** Telomerization of isoprene and CO₂ towards "COOIL" and polymerization of the corresponding lactone.

Telomerization of substituted 1,3-dienes like isoprene, which contains an internal methyl group, has been challenging due to reduced reactivity from increased steric hindrance[7]. Although some telomerization reactions with alcohols and amines have been reported[29–33], telomerization with $CO_2$ remains an outstanding challenge. In 1995, Dinjus and Leitner reported the isolation of two lactone isomers from isoprene and $CO_2$ in 1% yield (TON = 14) by preparatory thin-layer chromatography (pTLC)[34], in sharp contrast to up to 86% yield achieved with butadiene (Fig. 1b)[16–19]. In addition, achieving selectivity over several other lactone, acid, and terpene products, as well as controlling the formation of up to 4 regioisomers while maintaining high turnover numbers, complicates the reaction development[31].

In 2014, our group showed that a mixture of isoprene and butadiene could result in a poly-lactone material via a one-pot two-step process, though the isoprene content was not quantified[7]. These materials exhibited a lower glass transition temperature ($T_g = 63\,°C$) than poly-EVP ($T_g = 178–192\,°C$), which only consisted of butadiene-derived monomers, possibly due to the introduced methyl groups. Further lowering the $T_g$ of these isoprene-derived materials would impart them with favorable properties distinct from poly-EVP. Clearly, further catalyst development is required to establish an effective pathway for lactone production from cheap and available isoprene and $CO_2$ to tap into the vast potential of the isoprene economy.

In this study, we present a palladium-based telomerization catalyst for isoprene and $CO_2$, generating $CO_2$-Isoprene-Lactone "COOIL" (**1**) with 24 wt% $CO_2$ with turnover numbers >100 (Fig. 1c). The success of this reaction was dependent on a combination of a palladium(II) precursor, a triarylphosphine ligand, tetrabutylammonium acetate, and co-catalytic amounts of water. The resulting lactone isomers could be converted into low-molecular-weight polymers under Lewis acid-promoted conditions with the lactone units intact.

## Results and discussion

We initiated developing the telomerization reaction by screening phosphine and amine ligands in the presence of 0.07 mol% Pd(acac)₂ in acetonitrile under 2 MPa of $CO_2$ pressure (see Supplementary Information (SI) for details). All evaluated ligands resulted in low isoprene conversion and predominantly afforded mono- and sesquiterpenes without $CO_2$ incorporation, as determined by GC/MS analysis. Using the diamine ligand TEEDA, the desired δ-lactone **1** was generated with high selectivity, albeit only at trace amounts (0.1% isolated yield based on isoprene). Two of four possible regioisomers, **1a** and **1b**, were observed in a 1:3 ratio, with isomer **1b**, featuring an α-oxygen quaternary center, as the predominant isomer (Fig. 1c). The same two isomers were detected by Dinjus and Leitner in earlier work, albeit with the opposite regioselectivity[34].

The use of tetrabutylammonium acetate (TBAAc) as an additive, as reported by Bao et al.[35] enhanced the formation of the desired lactone products, reaching 2% yield (as determined by GC) while maintaining good selectivity (50–70% over other isoprene-derived products by GC). The hygroscopic reagent TBAAc readily absorbed ambient moisture during experiment preparation, which was mitigated by prolonged drying of the assembled autoclave under vacuum before charging the liquid reagents and conducting the reaction. Increasing the catalyst loading to 0.25 mol% and extending the reaction time allowed the isolation of **1** in 5% yield (TON = 19) in a regioisomer ratio (r.r.) of 1:11, favoring isomer **1b**.

With these initial results in hand, we further optimized the conditions to increase the yield of lactone **1**. Reducing the catalyst loading to 0.10 mol% and adding PPh₃ improved the yield to 12% and the lactone selectivity to 69% with a r.r. of 1:3 (Table 1, entry 1). Using the slightly more electron-rich ligand P(p-Tol)₃ further boosted the yield to 17% (**1**: 2.0 r.r.) (entry 2). In contrast, other phosphine and NHC ligands previously reported for the telomerization of butadiene with $CO_2$ were ineffective (entry 3 and SI). Increasing $CO_2$ pressure to 4 MPa marginally increased the yield (entry 5), while further adjustments to catalyst loading or temperature were ineffective (see SI). Testing different palladium precursors revealed that Pd(acac)₂ was optimal, with other Pd(II) complexes yielding less product (entries 6–7). Meanwhile, a Pd(0) precursor was incompatible with this reaction (entry 8).

Ammonium salts, in conjunction with a reductant, are known to promote the formation of nanoparticles from Pd(II) precursors[36]. Wang et al. recently reported that in situ generated Pd nanoparticles from Pd(acac)₂ and TBAAc are highly active in the telomerization of butadiene and $CO_2$, achieving yields comparable with homogeneous protocols[18,35]. We investigated whether nanoparticles were the active catalyst in our transformation. However, using colloidal Pd nanoparticles synthesized from Pd(acac)₂ and anhydrous TBAAc[37], no reaction occurred, irrespective of the presence of additional TBAAc and phosphine (entry 9, SI). In addition, using heterogenized Pd on ceria support[38], or in the form of a palladium-containing perovskite[39], resulted in no conversion (entries 10–11). These findings suggest that the active species are likely monomeric or small aggregates of homogeneous palladium species. We generally obtained homogeneous clear solutions after the reaction was completed, indicating no Pd black formation.

Having established that the combination of Pd(acac)₂, TBAAc, and P(p-Tol)₃ presents the optimal catalyst system for the isoprene telomerization, we addressed the variable moisture content due to the hygroscopic nature of TBAAc. Anhydrous TBAAc absorbed significant amounts of water when exposed to ambient air (25 °C, 60% relative humidity) for two hours, as discernable by IR and ¹H NMR

## Table 1 | Initial optimization of the telomerization of isoprene and CO$_2$

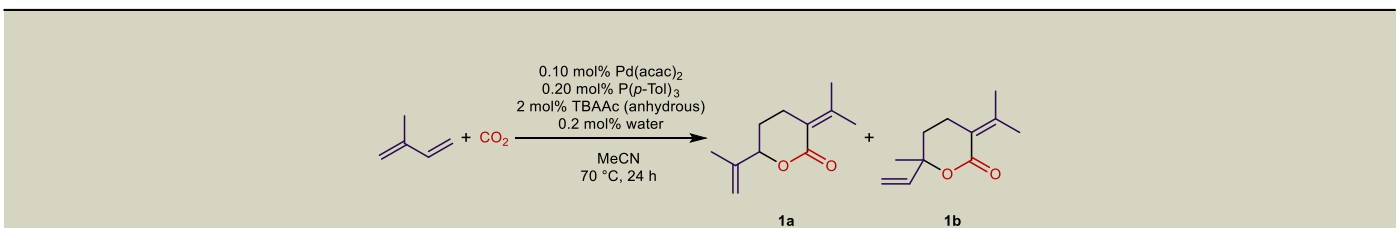

| Entry[a] | Deviation from optimal | Selectivity 1 (%)[b] | Yield (%)[c] | Ratio 1a:1b |
|---|---|---|---|---|
| 1 | PPh$_3$ | 69 | 12 | 1:3 |
| 2 | None | 78 | 17 (15)[d] | 1:2.0 |
| 3 | P(o-OMe-Ph)$_3$ | 30 | 2 | 1:10 |
| 4 | 80 °C | 60 | 8 | 1:3 |
| 5 | 4 MPa CO$_2$ | 81 | 19 | 1:2.0 |
| 6 | Pd(OAc)$_2$ | 56 | 11 | 1:4 |
| 7 | Pd(MeCN)$_2$Cl$_2$ | 61 | 8 | 1:2.4 |
| 8 | Pd$_2$(dba)$_3$ · CHCl$_3$ | 7 | <1 | 1:0.6 |
| 9 | Pd nanoparticles | n.d. | 0 | n.d. |
| 10 | Pd/CeO$_2$ | n.d. | 0 | n.d. |
| 11 | LaFe$_{0.95}$Pd$_{0.05}$O$_3$ | n.d. | 0 | n.d. |
| 12 | Anhydrous TBAAc | n.d. | 0 | n.d. |
| 13 | Anhydrous TBAAc exposed to air for 15 min | 59 | 9 | 1:3 |

[a]Reaction conditions: isoprene (25 mmol), Pd(acac)$_2$ (0.10 mol%), P(p-Tol)$_3$ (0.20 mol%), TBAAc (2.0 mol%), MeCN (2.5 mL), CO$_2$ (2 MPa), 70 °C, 24 h. [b]Determined by GC-FID area ratio of 1 versus all isoprene-derived products. [c]GC-FID yield with external standard n-decane. [d]Isolated yield.

## Table 2 | Control experiments of the telomerization of isoprene and CO$_2$ with well-defined water content

| Entry | Deviation from optimal | Selectivity 1 (%)[b] | Yield (%)[c] | Ratio 1a:1b |
|---|---|---|---|---|
| 1 | Without water | n.d. | 0 | n.d. |
| 2 | None | 77 | 19 | 1:1.8 |
| 3 | without TBAAc | n.d. | 0 | n.d. |
| 4 | without P(p-Tol)$_3$ | 46 | 2 | 1:12 |
| 5 | B$_2$pin$_2$ instead of TBAAc and water | n.d. | 0 | n.d. |
| 6 | 60 °C, 48 h | 81 | 10 | 1:3 |
| 7 | 60 °C, 96 h | 80 | 23 | 1:2.6 |
| 8 | 50 °C, 96 h | 80 | 20 | 1:2.0 |

[a]Reaction conditions: isoprene (25 mmol), Pd(acac)$_2$ (0.10 mol%), P(p-Tol)$_3$ (0.20 mol%), TBAAc (2.0 mol%), water (0.20 mol%), MeCN (2.5 mL), CO$_2$ (4 MPa), 70 °C, 24 h. [b]Determined by GC-FID area ratio of 1 versus all isoprene-derived products. [c]GC-FID yield with external standard n-decane.

measurements (see Fig. S6-9, SI). We rigorously dried the previously used commercial TBAAc batch by prolonged heating at 80 °C under vacuum, removing the bound water, as corroborated by IR measurements. Remarkably, using this anhydrous TBAAc no trace of **1** but only terpenes were produced (entry 12). Notably, when anhydrous TBAAc was allowed to age under ambient conditions for 15 min (20 °C, 80% rel. humidity), **1** was formed in 9%, implicating the beneficial effect of water in the reaction (entry 13).

To determine the optimal water amount for the reaction, we subsequently employed anhydrous TBAAc and added defined amounts of degassed water to the catalyst solution under an inert atmosphere. Without added water, the reaction produced only terpene products (Table 2, entry 1). Conversely, with the addition of

0.2 mol% of water (entry 2), equivalent to twice the amount of palladium species, we reproduced the optimal result with hydrated TBAAc (Table 1, entry 2). Thus, we obtained an isolated combined yield of 19% (TON = 93) of **1a** and **1b** and 77% selectivity. Omission of ligand or acetate shut down the reaction (entries 3–4). Wei and co-workers reported that a base is necessary to generate well-defined Pd(0) species and found that TBAAc, in conjunction with co-stoichiometric amounts of water, was more effective than other bases[40]. They also found that B$_2$pin$_2$ to be a more efficient reductant to produce Pd(0) species within short times. However, in this reaction, substituting TBAAc/water by B$_2$pin$_2$ did not promote the telomerization (entry 5). We could further decrease the reaction temperature to 50 °C by extending the reaction time (entries 6–8), achieving a yield of 23%

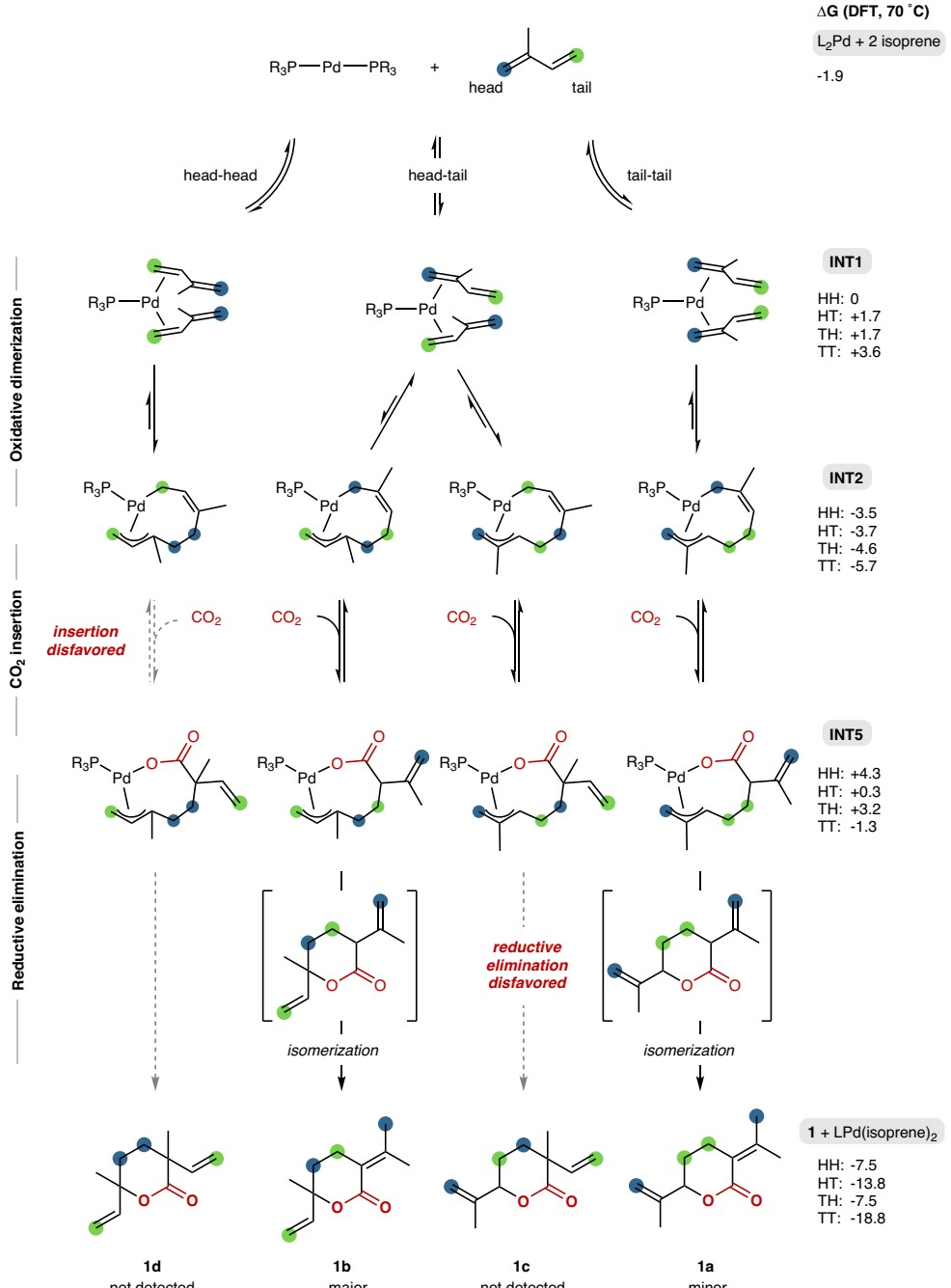

**Fig. 2 | Simplified mechanistic pathway for the palladium-catalyzed telomerization of isoprene and CO₂.** Solid arrows indicate a reactive path between intermediate, and dashed arrows indicate an unreactive one.

(TON = 114), the highest in this work, which is a magnitude larger than state-of-the-art (1%, TON = 14)[34]. The conversion of isoprene, as determined by [1]H NMR analysis, lied in the range of 50–60%. The optimized conditions were scalable, enabling a gram-scale synthesis of a mixture of **1a** and **1b** (1.63 g in total using 0.125 mmol of Pd, see SI).

Thus, water has a potential effect on the formation of lactone **1** over terpenes, although its exact role remains unclear. The beneficial effect of trace moisture in palladium-catalyzed coupling reactions has been well documented[40–44], including examples of the telomerization of butadiene/$CO_2$[6]. We hypothesize that a small amount of water might facilitate the reduction of the Pd(II) species to active Pd(0) by concurrent phosphine oxidation. Supporting this, we detected the corresponding phosphine oxide after the reaction by GC analysis. While reduction of Pd is generally assumed to occur readily in the presence

of phosphine ligands and trace moisture[41,45,46], the fate of the presumed Pd(0) species remains unexplored in the telomerization literature.

Beyond solely mediating precatalyst reduction, water appears to play other beneficial roles in this reaction. For instance, using a diborane as alternative reductant did not result in any catalytic activity (Table 2, entry 5). One potential additional role of water as a hydrogen bonding donor is increasing solubility and/or activating $CO_2$, as has been widely documented in the contexts of $CO_2$ adsorbents[47–49].

To gain insight into the mechanism of this transformation, we performed density functional theory (DFT) calculations on the pathways leading to the four possible lactone isomers. The proposed catalytic cycle consists of three principal stages: oxidative dimerization of

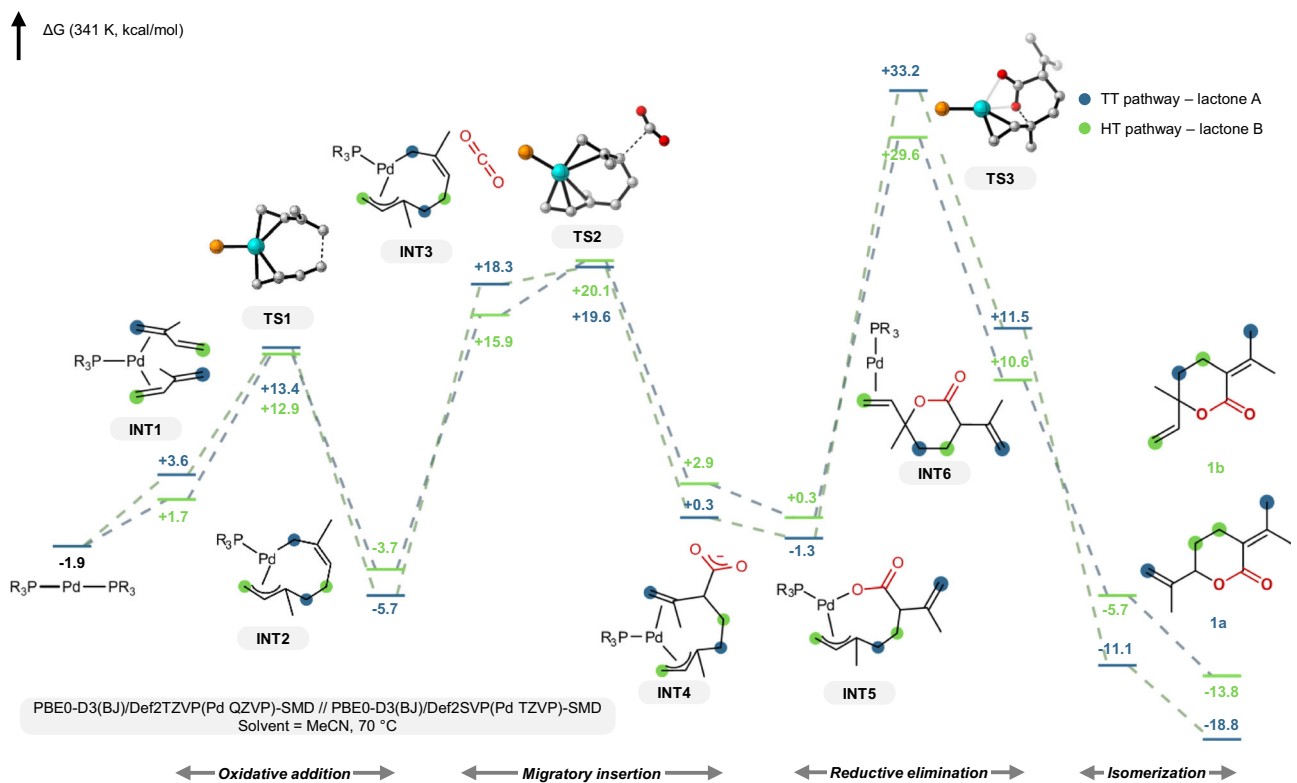

**Fig. 3 | Calculated free energy diagram, showing the HT and TT pathways leading to lactone products 1a and 1b, respectively.** Structures of the head-tail (HT) pathway are shown. Hydrogens and the aryl groups of the phosphine ligand are omitted for clarity. For the other pathways see SI.

## Table 3 | Polymerization conditions

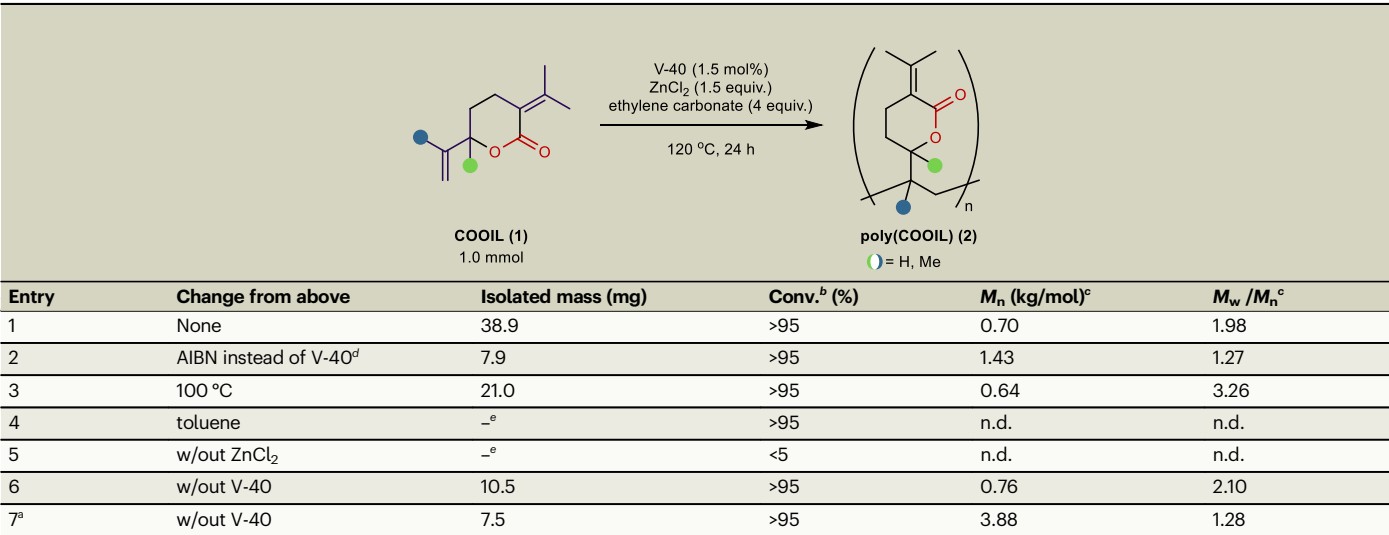

| Entry | Change from above | Isolated mass (mg) | Conv.[b] (%) | $M_n$ (kg/mol)[c] | $M_w$ /$M_n$[c] |
|---|---|---|---|---|---|
| 1 | None | 38.9 | >95 | 0.70 | 1.98 |
| 2 | AIBN instead of V-40[d] | 7.9 | >95 | 1.43 | 1.27 |
| 3 | 100 °C | 21.0 | >95 | 0.64 | 3.26 |
| 4 | toluene | –[e] | >95 | n.d. | n.d. |
| 5 | w/out ZnCl₂ | –[e] | <5 | n.d. | n.d. |
| 6 | w/out V-40 | 10.5 | >95 | 0.76 | 2.10 |
| 7[a] | w/out V-40 | 7.5 | >95 | 3.88 | 1.28 |

[a]0.50 mmol scale [b] Conversion was determined by [1]H NMR spectroscopy. [c] Determined by GPC in CHCl₃ with RI detector. MW was calculated using PS as standards. [d]V-40 = 1,1'-azobis(cyclohexane-1-carbonitrile). [e]no precipitated material.

two isoprene units, migratory insertion of $CO_2$, and reductive elimination of the C−O bond to form the lactone products (Fig. 2). Finally, outer-cycle isomerization leads to the more stable conjugated lactone products **1a** and **1b**.

The free energy diagram of the two predominant pathways, head-to-tail (HT) and tail-to-tail (TT), is depicted in Fig. 3. The two isoprene units can each coordinate in head- or tail-fashion, leading to three distinct Pd(0) diolefin complexes (**INT1**) by symmetry[32,50–52]. Oxidative dimerization (**TS1**) proceeds with a relatively low barrier resulting in

four possible palladacycles (**INT2**), and exists in a fast pre-equilibrium prior to the subsequent catalytic steps. The head-to-head (HH) dimer was found to have both the lowest transition state barrier and ground state energy due to reduction of steric repulsion, while the TT transition state (TS) barrier was the highest. In all cases a η¹-η³-coordination mode of the isoprene dimer was preferred to η³-η³-mode to relieve ring strain as previously found.

Outer-sphere approach by $CO_2$ is succeeded by migratory insertion and C-C bond formation (**INT3**). The initial approach is calculated

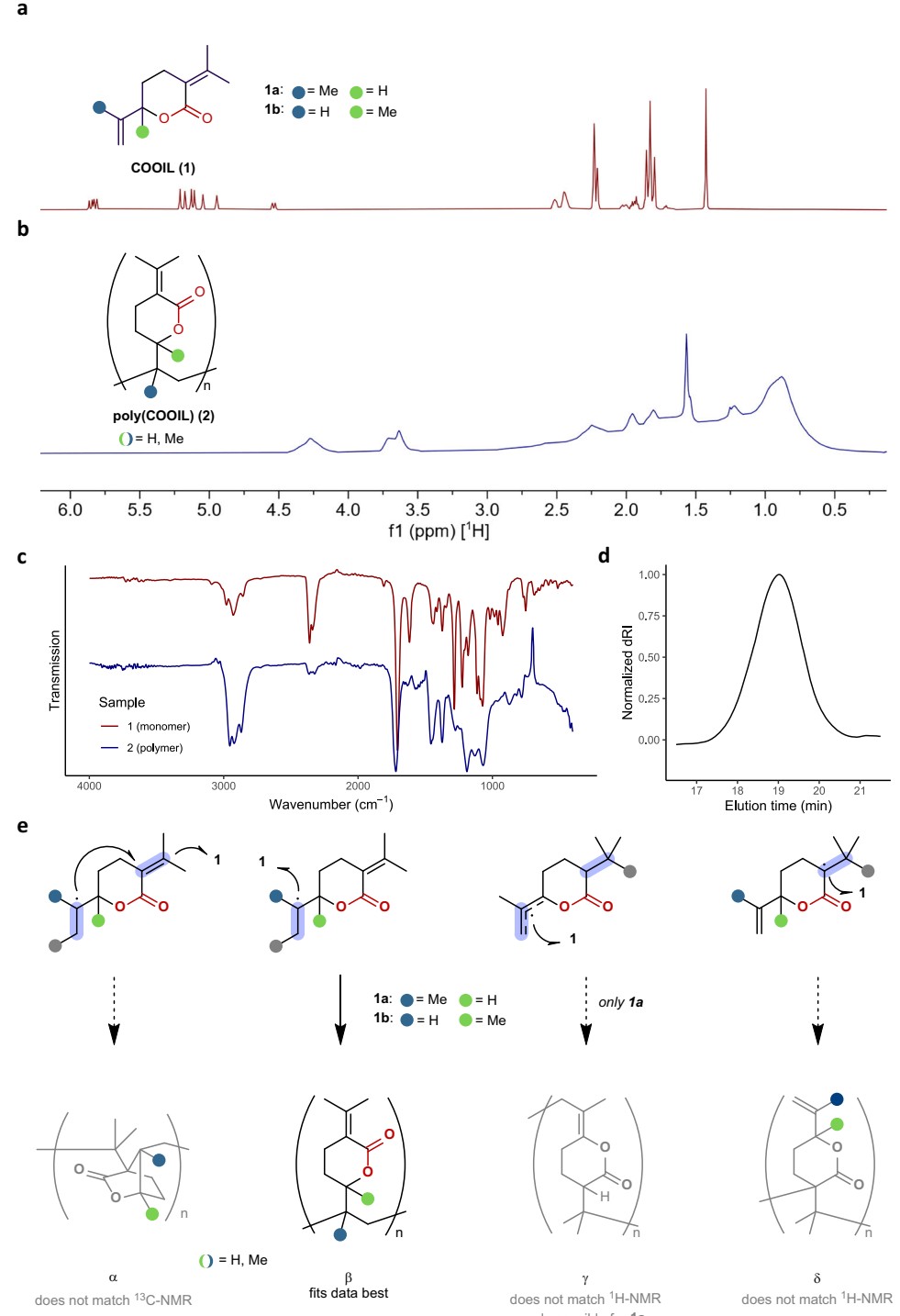

**Fig. 4 | Characterization of monomer 1 and polymer 2. a** [1]H NMR spectrum of lactone **1** (500 MHz, CDCl₃). **b** [1]H NMR spectrum of **2** (500 MHz, CDCl₃). **c** FTIR spectrum of **2**. **d** SEC trace of **2**. **e** Possible polymer structures.

to be energetically uphill under standard conditions, and the insertion transition states (**TS2**) are reactant-like. Notably, insertion into the HH palladacycle leads to a comparably less stable intermediate, which rationalizes the experimental absence of lactone **1d**. Conversely, the post-insertion intermediates (**INT4**) are low in energy for the HT, tail-to-head (TH) and TT pathways.

Following $CO_2$ insertion, rotation of the carboxylate sets up the reductive elimination step (**INT5**). The C-O reductive elimination (**TS3**) to furnish the lactone exhibits the highest transition state barriers among all elementary steps in the catalytic cycle, ranging from

29.6 kcal/mol for the HT pathway to 33.2 kcal/mol for the TT isomer to 37.4 kcal/mol for the TH isomer. This high energy barrier for ring closure is consistent with the moderate overall conversion observed experimentally, indicating that reductive elimination is the overall rate-limiting step. Further, the lack of formation of the TH-pathway derived lactone **1c** can be rationalized by the high barrier of reductive elimination compared to the other isomers. The reaction is thermo-dynamically favorable overall upon ligand exchange and release of the lactone product (ΔG HT: −5.7 kcal/mol; ΔG TT: −11.1 kcal/mol). Finally outer-cycle isomerization of the double bond generates the

experimentally observed lactones **1a** ($\Delta G = -16.8$ kcal/mol) and **1b** ($\Delta G = -11.9$ kcal/mol).

The product regioselectivity is governed by the relative free energy barriers of the reductive elimination transition states leading to the different lactone isomers and the relative thermodynamic stabilities of the final lactone products. Specifically, our calculations with the Pd/P($p$-Tol)$_3$ catalyst system show that the pathway forming lactone **1b'** (derived from head-to-tail coupling) is kinetically favored due to a lower reductive elimination barrier compared to the pathway forming **1a'** (derived from TT coupling). However, lactone **1a'** is calculated to be the thermodynamically more stable product. This is in alignment with experimental observations showing that the selectivity ratio of **1a** relative to **1b** increases with increasing reaction conversion (cf. Table 2, entries 2 versus 4 and SI). Conversely, the product distribution follows Curtin-Hammett conditions: it is governed by the relative barriers of the reductive elimination transition states, but not by the relative stabilities or formation rates of the initial dimeric intermediates.

Given the diverse polymerization pathways available for butadiene-derived EVP[3,4,53], we explored the potential of converting isoprene-derived monomer **1** into macromolecules with high $CO_2$ content. Unlike butadiene-derived EVP, **1** showed no reactivity with a free radical initiator, in all attempts under neat conditions leading to monomer recovery. This diminished reactivity of **1** can be rationalized by the tetrasubstituted double bond, that is even more sterically hindered than the tiglate moiety in butadiene-derived EVP. Similarly, emulsion polymerization and organocatalytic ring-opening polymerization protocols resulted in no conversion.

However, when ZnCl$_2$ as a promoter in ethyl carbonate (EC) solvent was employed in addition to V-40 as a radical initiator, full conversion of **1** into a polymer was observed according to $^1$H NMR analysis (Table 3, entry 1). Size-exclusion chromatography (SEC) analysis revealed that the material obtained under these conditions was oligomeric ($M_n = 0.7$ kg/mol, $M_w/M_n = 1.98$). Employing AIBN increased $M_n$ and reduced dispersity, but at the same time, reduced the amount of recovered material (entry 2). Raising the reaction temperature reduced control over the polymerization (entry 3), while apolar solvents resulted in poor material recovery (entry 4). As noted, ZnCl$_2$ was essential to initiate polymerization (entry 5). Interestingly, polymerization still occurred in the absence of a radical initiator (entry 6). On a smaller scale, polymer **2** with higher molecular weight and lower polydispersity could be isolated and characterized ($M_n = 3.9$ kg/mol, $M_w/M_n = 1.28$) (Table 3, entry 7 and Fig. 4d). The glass-transition temperature ($T_g$) of the obtained material was 44 °C, in line with the trend that the addition of methyl substituents significantly decreases $T_g$.

The repeating unit of **2** was deduced using IR and NMR spectral data (Fig. 4a–c) and by comparison with butadiene-derived poly-EVP and reference compounds (Fig. 4e, see SI for details). The $^{13}$C NMR chemical shift of the carbonyl at 169 ppm, along with the IR data rule out bicyclic structure α. In the $^1$H NMR spectrum, the signal at 4.3 ppm is assigned to the α-alkoxy proton in structure β arising from polymerization of the allylic ester in **1a** ($R^2 = H$). For **1b**, all signals can be assigned in the aliphatic region. Structures γ and δ can be ruled out as well, both from NMR data and because polymerization of tetrasubstituted olefins is generally disfavored[7,54].

In summary, we have achieved the catalytic telomerization of isoprene with $CO_2$, two of the most abundant carbon-containing molecules in the atmosphere. This accomplishment overcomes the long-standing challenge of telomerization of substituted 1,3-dienes with $CO_2$. The key to achieving consistent catalyst performance was selecting a suitable palladium/ligand combination and controlling the amount of co-catalytic water in the reaction.

Given the tremendous market potential of valorizing the abundant, renewable, and underutilized carbon-feedstocks of isoprene and $CO_2$, our work is poised to stimulate further research into this sustainably sourced lactone and its potential applications of sustainable polymers.

## Data availability

Details about the materials and methods, experimental procedures, characterization data, and computational studies, are included in this article and its supplementary files. Cartesian coordinates of structures for DFT experiments are provided in a separate .xlsx file as Supplementary Data. All additional data are available from the corresponding author upon request.

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

## Acknowledgements

M.D.R.L. acknowledges that this work was supported by the Swiss National Science Foundation (Postdoc.Mobility P500PN_ 214255). The computational studies were performed using resources of the Research

Center for Computational Science, Okazaki, Japan (Project: 22-IMS-C017).

## Author contributions

M.D.R.L.: Conceptualization, Methodology, Investigation, Formal Analysis, Writing—Original Draft, Funding acquisition. F.K.: Methodology, Investigation, Writing—Review & Editing. K.M.: Validation, Review & Editing. K.N.: Conceptualization, Supervision, Writing—Review & Editing.

## Competing interests

The authors declare no competing interests.
