## [Transparent Peer Review file · Nature Communications]

Gram-Scale Selective Telomerization of Isoprene and CO₂ Toward 100% Renewable Materials

Corresponding Author: Professor Kyoko Nozaki

Version 0:

Reviewer comments:

Reviewer #1

(Remarks to the Author)

This manuscript by Nozaki and coworkers is the first “modern” attempt to tackle the telomerization of CO₂ with isoprene rather than butadiene, generating new lactones. This is a very timely and important topic: related butadiene-derived lactones have absolutely exploded in importance/utility over the last 3 years, and translating that chemistry to (potentially) bioderived isoprene would be a major advance. Furthermore, initial attempts from Behr and Dinjus on this reaction decades ago cast a very long shadow on the area: this looked like a potential “unsolveable” problem. The report here is a good first step toward a solution accessing TON values exceeding 100 (albeit still at low yields and selectivities). The catalysis optimization focuses on the role of water, which to this reviewer still remains a bit ambiguous; and, the results are still all rather empirical. Given the scope of the problem, this is perhaps not surprising and doesn’t detract from the importance in this reviewer’s mind.

Table 1 entry 4: the 1a:1b ratio is written as 3:1, but I believe it should be 1:3.

Selectivity and yield are presented, but what is the conversion of the system? i.e. is there unreacted isoprene that can be recycled? Relatedly, if conversion < 100%, do longer reaction times lead to higher yields or is the catalyst system dead?

Can the catalyst system be reused?

I think in-text comparison to the Dinjus system (TON = 14) is warranted. While the original paper only obtains 1% yield, the lactones are formed catalytically and the advances here (yield, TON) should be put into that context.

The authors rather convincingly demonstrate that having some water in the reaction mixture is important, but I am less convinced that this is related to reduction of Pd(II) to Pd(0)—for example, using Pd₂(dba)₃, a Pd(0) source, still results in very poor catalysis, and using B₂Pi₂ as a reductant also fails. These reactions are a soup of additives, which makes systematic analysis difficult. But I think alternate explanations should at least be presented. For example, water can certainly act as a hydrogen-bonding bridge to facilitate all sorts of fundamental organometallic steps—what if it’s something like H-bonding to activate an inserted carboxylate toward reductive elimination?

I think Figure 2 and the related computational discussion could benefit from some expansion and clarification. First off, the oxidative coupling on Pd(0) is reversible for the formation of all 3 diallyl complexes according to the Carbo paper—this needs to be drawn, because it’s critical for the Curtin-Hammett-type explanation for the formation of 1a/1b. This is really the key observation: if it weren’t reversible, the CO₂ insertion wouldn’t ever happen. Further, inclusion of the dG values for each intermediate would be helpful, as well as calculation (and inclusion of the dG values) of all 4 possible lactone products (I agree that A is almost certainly the global thermodynamic product but knowing where C and D are in comparison would be instructive and simple to include). I was a little surprised that no effort was made to carry out DFT analysis of the Pd/Pt₂O₃ system, including simple single-point calculations of the oxidative dimerization products. Furthermore, the authors should be able to provide an approximation for ddG of the 2 productive CO₂ insertion pathways given they know Keq between oxidative dimerization products and the final product distribution.

Unfortunately, the polymerization of these isoprene-derived lactones isn’t very good. I had a couple of ideas that may be worth commenting on:

- did the authors consider attempting Lin's conditions for (aerobic) polymerization?
- the authors propose that the tetrasubstituted a,b-unsaturated alkene is likely a problem for propagation—agreed. Can the authors observe an end group by NMR (these are low molecular weight materials) that would be expected for formation of a (stabilized) tiglate radical that can't react further? Alternately, if some of this isoprene lactone monomer is included in an EVP polymerization which typically behaves much better, does it *inhibit* that polymerization reaction? MALDI-TOF of such a copolymerization may also reveal end-capping if so.

I found the labeling/assignment of the lactone mixture in Figure S11/S12 difficult to follow because the colors are all kind of similar, and the shapes the same. The integral/multiplicity box labels are also a different color than the chemdraws.

Reviewer #2

(Remarks to the Author)

The communication "Gram-Scale Selective Telomerization of Isoprene and CO₂: Toward 100% Renewable Materials" describes the catalytic conditions for producing sterically congested unsaturated lactones and their polymerization via radical propagation. While the coupling of butadiene and CO₂ is well established at pre-pilot scales, the comparable reactivity with isoprene is less developed and could have a significant impact on the development of sustainable polymers. Specifically, the isoprene reaction is challenging due to the low conversions and unfavorable CO₂ insertion which results in isoprene oligomers without CO₂ insertion. The merits of this work include the discovery that precise moisture content and ligand selection can significantly improve the selectivity of the isoprene telomerization for CO₂ incorporation. Although the overall yields remain low (15%) this is a significant improvement to the prior art (~2). Another limitation of the work is that unlike the butadiene/CO₂ derived lactone this monomer only affords low molecular weight polylactones (<5,000 g/mol) when subjected to radical polymerization and the resulting material properties are only briefly investigated.

Several detailed points the authors should address before publication are:

- 1) Overall the author's data supports their conclusions and is conducted with scientific rigor. However several parts of the title were judged misleading by this reviewer. While "gram-scale" is accurate, the largest reaction afforded 1.6g of a monomer mixture. Any materials platform will require significantly more productive system than 114 turnovers with a precious metal catalyst. Additionally, bio-isoprene produces more greenhouse gases, requires more energy, and consumes more water than fossil-isoprene which significantly detracts from the authors assertion of "100% renewable materials." [see <https://doi.org/10.1021/acssuschemeng.2c05764>] And lastly, "selectivity" was achieved by suppressing the formation of terpene coproducts, but the reaction is far from achieving high conversion. These criticisms notwithstanding, the work is a critical breakthrough to accessing these lactones for the first time. Perhaps the authors can provide a reasonable response to these points and revise language asserting "high turnover numbers" and include a more quantitative discussion on the observed conversion values.
- 2) The authors introduce the concept that isoprene-derived monomers would impart favorable T_g properties, but never discuss the T_g in the main text. Figure S24 reports the "DSC chart of CO₂/butadiene copolymer 2", but the transitions are not clearly defined. Is there phase separation or polymer degradation? The TGA figure S23 should be revised to include a plot of wt% remaining vs temperature, so it may be more readily interpreted by readers.
- 3) Fig S10 single electron arrows should be corrected. Although the arrows are simplified for convenience, they should accurately reflect radical mechanisms. Including the transannular hydrogen atom transfer for the gamma microstructure.
- 4) Section S1.2 when listing the ¹H NMR peaks for isomer 1a there is a total of 17 protons listed while only 16 are present in the molecules. Double check the integrations reported.
- 5) The authors describe investigating "Cationic ring opening polymerization" conditions, but appear to report base-catalyzed TBD systems, which would be better characterized as "organocatalytic ROP".
- 6) Typographical error: Several supplemental figures were labeled as "CO₂/butadiene" when it appears to be reporting "CO₂/isoprene" compounds and should be double checked throughout (NMRs, Fig S22, S23, S24).
- 7) Typographical error: Section S5 "Computational details" reference 17 and 18 should be superscripted.
- 8) Typographical error: Table S6 entry 6 contains a period instead of indicating a complex

Version 1:

Reviewer comments:

Reviewer #1

(Remarks to the Author)

The authors have done an admirable job addressing the critiques of both reviewers--I think the inclusion of the more detailed DFT analysis adds a significant amount of insight into this work for future development. I am in favor of publication with a couple of very minor corrections:

- inclusion of some of the energy values into figure 2 would enhance readability
- the "curved" lines connecting the reaction coordinate diagram in figure 3 make it a little bit hard for this reviewer to visually compare energy levels between T_Ses and INTs: the eye is typically drawn better to straight lines when making comparisons. This is a stylistic point, but I'd encourage just using straight lines for the reaction coordinate diagrams.

- I think the authors misunderstood my comment about the co-polymerization of the isoprene-derived lactone with the butadiene-derived one. While it's certainly true that they demonstrated the copolymerization in their initial 2014 report, this was with the hetero-coupled isoprene/butadiene, which may not necessarily have the same issues as a full-isoprene lactone (which can't avoid forming highly substituted radicals, etc). Upon re-examining the complexity of the polymer spectra and seeing the MALDI, this reviewer thinks that including this experiment is NOT worth the time for this paper--but, something to think about toward the future.

Reviewer #2

(Remarks to the Author)

The authors have adequately revised their manuscript to more accurately represent their findings and conclusions. In addition, the typographical errors have been addressed to improve the clarity and experimental details. Although both the catalytic yield and the polymerization results should be improved, the work significantly advances and broadens the field of diene/CO₂ coupling and copolymerization.

Response to Decision Letter

Reviewer #1:

This manuscript by Nozaki and coworkers is the first “modern” attempt to tackle the telomerization of CO₂ with isoprene rather than butadiene, generating new lactones. This is a very timely and important topic: related butadiene-derived lactones have absolutely exploded in importance/utility over the last 3 years, and translating that chemistry to (potentially) bioderived isoprene would be a major advance. Furthermore, initial attempts from Behr and Dinjus on this reaction decades ago cast a very long shadow on the area: this looked like a potential “unsolveable” problem. The report here is a good first step toward a solution accessing TON values exceeding 100 (albeit still at low yields and selectivities). The catalysis optimization focuses on the role of water, which to this reviewer still remains a bit ambiguous; and, the results are still all rather empirical. Given the scope of the problem, this is perhaps not surprising and doesn’t detract from the importance in this reviewer’s mind.

— We thank the reviewer for their positive evaluation.

Table 1 entry 4: the 1a:1b ratio is written as 3:1, but I believe it should be 1:3.

— We apologize for our oversight. We have corrected the regioselectivity ratio in question.

Selectivity and yield are presented, but what is the conversion of the system? i.e. is there unreacted isoprene that can be recycled? Relatedly, if conversion < 100%, do longer reaction times lead to higher yields or is the catalyst system dead?

— We determined the conversion of isoprene by ¹H NMR analysis of the reaction mixture after release of CO₂ at 0 °C, taking precautions to minimize isoprene loss. Typical isoprene conversion achieved in our system is in the range of 50-60%. The following ¹H NMR spectrum was recorded from a reaction sample after venting CO₂. 1,3,5-Trimethoxybenzene was added as an internal standard to quantify the isoprene amount remaining. The conversion was calculated to be 60%.

This indeed means a significant amount of unreacted isoprene remains after the reaction. We added a sentence mentioning the conversion into the manuscript results section.

Regarding the effect of reaction time and catalyst turnover number, we observed that extending the reaction time does lead to higher isoprene conversion and product yield, as shown in Table S11 in the Supporting Information. However, attempts to take aliquots for *operando* isoprene conversion measurements and subsequently re-pressurize the system led to a loss of catalyst activity. We attribute this to sensitivity of the catalyst to moisture and oxygen that could not be completely excluded during these manipulations. In principle, a process involving continuous distillation of isoprene and lactone can be envisioned (and has been realized for butadiene by Behr and Heite) but requires further process research.

Can the catalyst system be reused?

— We thank the reviewer for their important question. Reusing the catalyst system in this specific reaction presents significant practical challenges. The lactone product is non-volatile, requiring vacuum distillation at 180 °C, which makes its separation from the solid catalyst difficult to achieve under the inert conditions required by our catalyst system.

Furthermore, when we subjected the isolated lactone to freshly prepared catalyst under nitrogen atmosphere, we observed a complex ¹H NMR spectrum which hinted at decomposition or isomerization of the lactone. This suggests that the lactone product is also reactive towards the catalyst under reaction conditions.

Due to the need to separate the non-volatile product from the catalyst, combined with the insert handling requirements, we were unable to demonstrate catalyst reuse under typical conditions. In theory it can be feasible, as Heite and Behr have demonstrated that in the scope of their miniplant publication.

I think in-text comparison to the Dinjus system (TON = 14) is warranted. While the original paper only obtains 1% yield, the lactones are formed catalytically and the advances here (yield, TON) should be put into that context.

— We thank the reviewer for their suggestion. We emphasized our result and put it into context with Dinjus and Leitner's work by adding the following details to the paragraph before Table 2 (new text underlined):

To determine the optimal water amount for the reaction, we subsequently employed anhydrous TBAAc and added defined amounts of degassed water to the catalyst solution under an inert atmosphere. Without added water, the reaction produced only terpene products (Table 2, entry 1). Conversely, with the addition of 0.2 mol% of water (entry 2), equivalent to twice the amount of palladium species, we reproduced the optimal result with hydrated TBAAc (Table 1, entry 2). Thus, we obtained an isolated combined yield of 19% (TON = 93) of **1a** and **1b** and 77% selectivity. Omission of ligand or acetate shut down the reaction (entries 3–4). Wei and co-workers reported that a base is necessary to generate well-defined Pd(0) species and found that TBAAc, in conjunction with co-stoichiometric amounts of water, was more effective than other bases.⁴³ They also found that B₂pin₂ to be a more efficient reductant to produce Pd(0) species within short times. However, in this reaction, substituting TBAAc/water by B₂pin₂ did not promote the telomerization (entry 5). We could further decrease the reaction temperature to 50 °C by extending the reaction time (entries 6–8), achieving a yield of 23% (TON = 114), the highest in this work, which is a magnitude larger than state-of-the-art (1%, TON = 14).³² The optimized conditions were scalable, enabling a gram-scale synthesis of a mixture of **1a** and **1b** (1.63 g in total using 0.125 mmol of Pd, see SI).

The authors rather convincingly demonstrate that having some water in the reaction mixture is important, but I am less convinced that this is related to reduction of Pd(II) to Pd(0)—for example, using Pd2(dba)3, a Pd(0) source, still results in very poor catalysis, and using B2Pin2 as a reductant also fails. These reactions are a soup of additives, which makes systematic analysis difficult. But I think alternate explanations should at least be presented. For example, water can certainly act as a hydrogen-bonding bridge to facilitate all sorts of fundamental organometallic steps—what if it's something like H-bonding to activate an inserted carboxylate toward reductive elimination?

— This reviewer correctly states that the presence of water can have several influences on the reaction, beyond palladium reduction. For instance, water is known to enhance the activation of CO₂ (*Chem. Rev.* 2021, **121**, 13, 7280–7345.). In addition, numerous examples of compounds bearing HO/HN groups are known as enhanced CO₂ absorber or catalyst (e.g. *Acc. Chem. Res.* 2024, **57**, 17, 2512–2521; *ACS Cent. Sci.* 2018, **4**, 3, 397–404; *ChemCatChem*, 2025, **17**, e202401394.). Another example are aqueous amine scrubbers which are used as CO₂ absorbers in the industry for decades (e.g. *Science*, 2009, **325**, 1652–1654; *Energy Procedia*, 2012, **23**, 45–54;). This supports the hypothesis

that water-mediated hydrogen bonding interactions can play a key role in the activation of CO₂.

Further, water might stabilize intermediates and transition states, and/or modify the solubility of CO₂ in the organic phase. We do not intend to speculate the role of water as the currently available data does not lead to clear conclusions. Significant further studies with spectroscopic experts would be called for to clarify this, which is beyond the scope of this work.

Computational treatment of hydrogen bond networks in organometallic contexts, such as this one, is highly complex and beyond the scope of this study. We foresee a future collaboration with specialists in computations to reach more insight into the mechanism and the role of water.

In conclusion, we have added the following paragraph to draw attention to other roles of water than acting as a reductant (new text underlined).

Thus, water has a potential effect on the formation of lactone **1** over terpenes, although its exact role remains unclear. The beneficial effect of trace moisture in palladium-catalyzed coupling reactions has been well documented,^{40,44-47} including examples of the telomerization of butadiene/CO₂.⁶ We hypothesize that a small amount of water might facilitate the reduction of the Pd(II) species to active Pd(0) by concurrent phosphine oxidation. Supporting this, we detected the phosphine oxide after the reaction by GC analysis. While reduction of Pd is generally assumed to occur readily in the presence of phosphine ligands and trace moisture,³⁸⁻⁴⁰ the fate of the presumed Pd(0) species remains unexplored in the telomerization literature.

Beyond solely mediating precatalyst reduction, water appears to play other beneficial roles in this reaction. For instance, using a diborane as alternative reductant did not result in any catalytic activity (Table 2, entry 5). One potential additional role of water as a hydrogen bonding donor is increasing solubility and/or activating CO₂, as has been widely documented in the contexts of CO₂ adsorbents.⁴³⁻⁴⁵

I think Figure 2 and the related computational discussion could benefit from some expansion and clarification. First off, the oxidative coupling on Pd(0) is reversible for the formation of all 3 diallyl complexes according to the Carbo paper—this needs to be drawn, because it's critical for the Curtin-Hammett-type explanation for the formation of 1a/1b. This is really the key observation: if it weren't reversible, the CO₂ insertion wouldn't ever happen.

— We thank this reviewer for their remark regarding the mechanistic proposal. We have conducted extensive DFT calculations and can now present the full catalytic pathway. As this reviewer pointed out correctly, the oxidative addition is reversible and follows Curtin-Hammett-type kinetics. We comment on the outcomes of the DFT study below. With a clarified mechanistic proposal in hand, we improved Figure 2 and its surrounding discussion. We show that oxidative cyclization is reversible and fast, and therefore has no bearing on the regioselectivity of the final products.

Further, inclusion of the dG values for each intermediate would be helpful, as well as calculation (and inclusion of the dG values) of all 4 possible lactone products (I agree that A is almost certainly the global thermodynamic product but knowing where C and D are in comparison would be instructive and simple to include). I was a little surprised that no effort was made to carry out DFT analysis of the Pd/Ptol3 system, including simple single-point calculations of the oxidative dimerization products. Furthermore, the authors should be able to provide an approximation for ddG of the 2 productive CO₂ insertion pathways given they know Keq between oxidative dimerization products and the final product distribution.

— We agree that DFT calculations are crucial for understanding this complex reaction and have conducted extensive DFT studies to investigate the reaction mechanism in detail. These comprehensive results are now included in the revised manuscript, adding a significant amount new data and discussion. In line with this, Figure 2 has been updated and Figure 3 depicting a free energy diagram has been added to the main text. Also, section 5 in the SI has been edited accordingly.

As suggested, we calculated the free energies ΔG for all four possible lactone products (**1a-d**) and provide them in Table S14 in the Supporting Information. We can confirm lactone **1a** as the thermodynamically most stable product, followed by **1b** and the direct reaction products **1a'** and **1b'** prior to double bond isomerization.

We further performed a detailed analysis of the potential energy surface of all four reaction pathways, beginning with the oxidative cyclization pathways, dubbed head-to-head, head-to-tail and tail-to-tail. These are in a fast pre-equilibrium prior to subsequent steps, as this reviewer correctly points out.

Our calculations reproduce the experimentally observed regioselectivity, showing that the overall barriers leading to lactones **1a** (via the tail-to-tail pathway) and **1b** (via the head-to-tail pathway) are significantly lower than those for potential lactones **1c** and **1d** (originating from the HH and TH pathways). Specifically, the lack of stabilization the intermediate post CO₂ insertion in the HH pathway, and reductive elimination in the TH pathway explain the experimental absence of lactones **1d** and **1c**. Furthermore, comparing the calculated reductive elimination barriers and product stabilities correctly

identifies lactone **1b** as the kinetically preferred product and lactone **1a** as the thermodynamically favored product, consistent with our experimental observations over time.

In conclusion, the computational results rationalize the origin of the experimentally observed regioselectivity and reveal that the reductive elimination step is the rate-limiting step in this transformation. The isoprene dimerization intermediates are described by Curtin-Hammett conditions which have no bearing on the final product distribution. Therefore, the unproductive but kinetically favorable HH dimerization product can dissociate to eventually form the thermodynamically more favorable lactones arising from the HT and TT pathways. While we have not included water in the calculations due to computational complexity and questionable accuracy, we acknowledge its presence might influence the migratory insertion and reductive elimination, as well as the solubility of CO₂ in solution.

Unfortunately, the polymerization of these isoprene-derived lactones isn't very good. I had a couple of ideas that may be worth commenting on:

- *did the authors consider attempting Lin's conditions for (aerobic) polymerization?*

— We thank the reviewer for their suggestions regarding the polymerization of lactone **1**. As shown in the SI in Table S12, we have attempted conditions similar to Lin et al. (*ACS Macro Letters* **2017**, 6 (12), 1373–1378). Specifically, when we heated neat monomer at 180 °C under air, we did not observe any polymerization but decomposition towards an unidentified small molecule. We reason that a radical is initially formed but is not able to propagate in the absence of a Lewis acid. We have added experimental details to Table S12 and cited Lin's work.

- *the authors propose that the tetrasubstituted a,b-unsaturated alkene is likely a problem for propagation—agreed. Can the authors observe an end group by NMR (these are low molecular weight materials) that would be expected for formation of a (stabilized) tiglate radical that can't react further?*

— We were not able to observe the end groups of the polymer by ¹H NMR because the spectrum featured significantly broadened signals (c.f. Fig. S15). On the other hand, the ¹³C NMR spectrum did not allow us to identify signals arising from the chain ends.

*Alternately, if some of this isoprene lactone monomer is included in an EVP polymerization which typically behaves much better, does it *inhibit* that polymerization reaction?*

— Our group has previously reported the co-polymerization of CO₂, butadiene and isoprene (*Nat. Chem.* **2014**, 6 (4), 325–33), albeit with low isoprene incorporation rate. Thus, in principle isoprene does not inhibit the polymerization but is far away from being statistically incorporated into the chain.

MALDI-TOF of such a copolymerization may also reveal end-capping if so.

— We thank the reviewer for their suggestion. We attempted collection of MALDI-TOF data on the polymer samples but obtained low signal to noise spectra that could not be evaluated. While the cause for the lack of well-defined repeating peaks is unclear, a contributing cause can be assumed to be significant cross-linking (c.f. *J. Am. Chem. Soc.* **2019**, 141, 10938–10942) which, even at low percentage, can cause a complicated spectrum.

I found the labeling/assignment of the lactone mixture in Figure S11/S12 difficult to follow because the colors are all kind of similar, and the shapes the same. The integral/multiplicity box labels are also a different color than the chemdraws.

— We thank the reviewer for their valuable suggestion. We have clarified the color palette for Figures S12 and S14 as shown below.

Figure S1. Assigned ^1H NMR spectrum (500 MHz, CDCl_3) of lactone mixture **1a** and **1b**.

Figure S2. Assigned $^{13}\text{C}\{^1\text{H}\}$ NMR spectrum (126 MHz, CDCl_3) of lactone mixture **1a** and **1b**.

Reviewer #2:

The communication “Gram-Scale Selective Telomerization of Isoprene and CO₂: Toward 100% Renewable Materials” describes the catalytic conditions for producing sterically congested unsaturated lactones and their polymerization via radical propagation. While the coupling of butadiene and CO₂ is well established at pre-pilot scales, the comparable reactivity with isoprene is less developed and could have a significant impact on the development of sustainable polymers. Specifically, the isoprene reaction is challenging due to the low conversions and unfavorable CO₂ insertion which results in isoprene oligomers without CO₂ insertion. The merits of this work include the discovery that precise moisture content and ligand selection can significantly improve the selectivity of the isoprene telomerization for CO₂ incorporation. Although the overall yields remain low (15%) this is a significant improvement to the prior art (~2). Another limitation of the work is that unlike the butadiene/CO₂ derived lactone this monomer only affords low molecular weight polylactones (<5,000 g/mol) when subjected to radical polymerization and the resulting material properties are only briefly investigated.

— We thank the reviewer for their overall positive assessment of the scientific rigor and data presented in our manuscript.

Several detailed points the authors should address before publication are: 1) Overall the author’s data supports their conclusions and is conducted with scientific rigor. However several parts of the title were judged misleading by this reviewer.

While “gram-scale” is accurate, the largest reaction afforded 1.6g of a monomer mixture. Any materials platform will require significantly more productive system than 114 turnovers with a precious metal catalyst.

— We also appreciate the critical comment regarding the interpretation of "gram-scale" in the title in the context of industrial readiness. We fully agree that the current catalyst system, while representing a significant advance, is not yet optimized or productive enough for immediate large-scale industrial synthesis, not mentioning an industrial process.

Our use of the term "gram-scale" in the title was intended to highlight a substantial step-up in reaction scale compared to previous studies that produced milligrams of isoprene-CO₂ derived lactones. Achieving >100 TON at the gram scale represents a significant milestone for this reaction, and allows to produce sufficient material for studies of EVP-like materials derived from this lactone. We hope our finding will stimulate further research to enhance the catalyst's productivity and stability to the point that an industrially relevant process can be conceived, and ultimately to reevaluate the potential of isoprene for sustainable materials production.

Additionally, bio-isoprene produces more greenhouse gases, requires more energy, and consumes more water than fossil-isoprene which significantly detracts from the authors assertion of “100% renewable materials.”

[see <https://doi.org/10.1021/acssuschemeng.2c05764>]

We appreciate the reviewer's comment regarding the current environmental footprint of bio-derived isoprene. We agree that, based on present technologies, its production can indeed involve higher greenhouse gas emissions, energy, and water use than optimized fossil routes. We added a note within the introduction of the main text to acknowledge these points, including a reference to Dunn et al.

We further would like to clarify that we aim to distinguish between a feedstock derived from a renewable source and the overall sustainability or economic viability of its current production process. Isoprene has the potential to be derived from renewable biomass or the atmosphere (cf. refs 22-23). Our use of "100% renewable materials" refers to the potential to be sourced from a renewable natural source, independent of the challenges currently associated with its large-scale, cost-effective, and sustainable production.

Our paper focuses on demonstrating a chemical transformation utilizing isoprene. The significance of our work lies in overcoming a decade-old dogma that isoprene is unproductive in the palladium-catalyzed telomerization with CO₂. While we acknowledge the reviewer's valid point about the question of sourcing isoprene sustainably, it does not diminish the relevance of demonstrating that actively researched EVP-like materials can be produced from feedstocks capable of being renewably sourced in the future.

In conclusion, we have added the above-mentioned reference and adjusted the paragraph regarding isoprene in the introduction as follows (new/modified text underlined):

Isoprene, a C₅ hydrocarbon, is the most abundantly emitted hydrocarbon in the atmosphere after methane.^{20,21} It is estimated that emissions release 400–500 teragrams of carbon (Tg C) from plants per year,^{22,23} approximately matching the amount of biogenic methane emissions.^{22,24} In contrast to butadiene, which is derived from fossil resources, isoprene is a feedstock with the inherent potential for renewable sourcing independent of petrochemical processes, albeit current production routes from biomass are not competitive in terms of energy utilization.²⁵ Despite its abundance and the versatility of its 1,3-diene motive, isoprene's use as a renewable carbon source is largely focused on natural rubber and adhesive production.^{26–28} This underutilization of such an abundant resource underscores the need to explore its potential for sustainable polymer production. Sequestering the two outstandingly abundant climate gases, CO₂ and isoprene,

into an EVP analog and its derived polymers could be a transformative step towards a more sustainable polymer economy.

And lastly, “selectivity” was achieved by suppressing the formation of terpene coproducts, but the reaction is far from achieving high conversion. These criticisms notwithstanding, the work is a critical breakthrough to accessing these lactones for the first time. Perhaps the authors can provide a reasonable response to these points and revise language asserting “high turnover numbers” and include a more quantitative discussion on the observed conversion values.

— We thank the reviewer for their positive evaluation and valuable suggestions. We have revised our language asserting high turnover numbers. We also conducted more experiments to clarify the conversion of isoprene. We quantified the conversion of isoprene by ^1H NMR analysis of the reaction mixture after release of CO_2 at $0\text{ }^\circ\text{C}$ to minimize loss of isoprene in the process. The conversion depends on the experimental run and lies roughly around 50-60%. We added a sentence mentioning the conversion into the manuscript results section.

The following ^1H NMR spectrum was recorded from a reaction sample after venting CO_2 . 1,3,5-Trimethoxybenzene was added as an internal standard to quantify the isoprene amount remaining. The conversion was calculated to be 60%.

2) The authors introduce the concept that isoprene-derived monomers would impart favorable T_g properties, but never discuss the T_g in the main text. Figure S24 reports the “DSC chart of CO_2 /butadiene copolymer 2”, but the transitions are not clearly defined. Is there phase separation or polymer degradation? The TGA figure S23 should be revised

to include a plot of wt% remaining vs temperature, so it may be more readily interpreted by readers.

— As suggested, we have now added a discussion of the T_g of the obtained polymer in the main text. In the SI we have also revised Fig. S23 (TGA) and S24 (DSC) for improved interpretability.

Figure S3. TG chart of CO₂/isoprene copolymer **2** measured at a heating rate of 10 °C/min from 30 °C to 515 °C. The y-axis represents normalized weight loss (black), and differential thermal analysis (DTA) (red).

Figure S4. DSC chart of CO₂/isoprene copolymer **2**.

Regarding the questions about transitions and potential phase separation or degradation by this reviewer, our polymer is an opaque material and visual inspection does not indicate macroscopic island/ocean type phase separation typical of block copolymers. The TGA data (Figure S23) shows the polymer exhibits high thermal stability up to approximately 200 °C, with decomposition occurring at higher temperatures. We acknowledge that precisely defining all transitions from the DSC is challenging, likely due

in part to the propensity of these poly(lactone)s to undergo cross-linking via ring-opening reactions, which can influence thermal analysis results. A detailed understanding of the material properties will require further dedicated studies beyond the scope of the present work.

3) Fig S10 single electron arrows should be corrected. Although the arrows are simplified for convenience, they should accurately reflect radical mechanisms. Including the transannular hydrogen atom transfer for the gamma microstructure.

— We drew the radical mechanism more explicitly in Fig. S10 to aid the reader, including the proposed HAT transfer intermediate leading to gamma structure. Also, we added more detailed explanations in the text paragraph preceding the figure (section 1.4.2).

Two direct addition/propagation products can be envisioned by attack of a radical on the allylic olefin (left pathway) or the α,β -unsaturated ester (right pathway) (Fig. S10). Both intermediates can further react by two pathways to afford up to four possible repeating units. Intramolecular cyclization of the α -oxygen radical onto the conjugated olefin would result in bicyclic structure α . Meanwhile intermolecular radical addition would result in structure β . In case of the enolate-like radical, hydrogen atom transfer would lead to a stabilized allylic radical that could further react to structure γ . Otherwise, structure δ might be generated.

Figure S5. Assignment of the microstructure of **2**. Spectral data of **S1**¹³ and **S2**.¹⁴

4) Section S1.2 when listing the ^1H NMR peaks for isomer 1a there is a total of 17 protons listed while only 16 are present in the molecules. Double check the integrations reported.

— We apologize for our oversight. The signal of isomer 1a at 2.20 (s, 2H) was invertedly reported as 3H.

5) The authors describe investigating “Cationic ring opening polymerization” conditions, but appear to report base-catalyzed TBD systems, which would be better characterized as “organocatalytic ROP”.

— We adjusted our description of the TBD systems to call them “organocatalytic ring-opening polymerization”.

6) *Typographical error: Several supplemental figures were labeled as “CO₂/butadiene” when it appears to be reporting “CO₂/isoprene” compounds and should be double checked throughout (NMRs, Fig S22, S23, S24).*

— We apologize for our oversight. We have corrected the figure captions in question.

7) *Typographical error: Section S5 “Computational details” reference 17 and 18 should be superscripted.*

— The two references that did not appear in the bibliography have been added.

8) *Typographical error: Table S6 entry 6 contains a period instead of indicating a complex*

— The notation was corrected to indicate a solvent complex.

Response to Decision Letter

Reviewer #1:

The authors have done an admirable job addressing the critiques of both reviewers--I think the inclusion of the more detailed DFT analysis adds a significant amount of insight into this work for future development. I am in favor of publication with a couple of very minor corrections:

— We thank the reviewer for their positive assessment of additional data presented in the revised manuscript.

- inclusion of some of the energy values into figure 2 would enhance readability

— Energy values of the depicted intermediates were added on the right pane.

Before revision:

After revision:

- the "curved" lines connecting the reaction coordinate diagram in figure 3 make it a little bit hard for this reviewer to visually compare energy levels between TSes and INTs: the eye is typically drawn better to straight lines when making comparisons. This is a

stylistic point, but I'd encourage just using straight lines for the reaction coordinate diagrams.

— We followed the reviewer's suggestion and replaced the Gaussian curves with straight lines.

Before revision:

After revision:

Accordingly, Figure 24 in the SI was also updated.

Before revision:

After revision:

• I think the authors misunderstood my comment about the co-polymerization of the isoprene-derived lactone with the butadiene-derived one. While it's certainly true that they demonstrated the copolymerization in their initial 2014 report, this was with the hetero-coupled isoprene/butadiene, which may not necessarily have the same issues as a full-isoprene lactone (which can't avoid forming highly substituted radicals, etc). Upon re-examining the complexity of the polymer spectra and seeing the MALDI, this reviewer thinks that including this experiment is NOT worth the time for this paper--but, something to think about toward the future.

— We thank this reviewer for clarifying their intent and acknowledge no further experiments are requested. We keep the suggested the co-polymerization of the

isoprene-derived lactone with the butadiene-derived one in mind for future research.

Reviewer #2:

The authors have adequately revised their manuscript to more accurately represent their findings and conclusions. In addition, the typographical errors have been addressed to improve the clarity and experimental details. Although both the catalytic yield and the polymerization results should be improved, the work significantly advances and broadens the field of diene/CO₂ coupling and copolymerization.

— We thank the reviewer for their overall positive evaluation.